# Self-Propulsion Enhances Polymerization

**DOI:** 10.3390/e22020251

**Published:** 2020-02-22

**Authors:** Maximino Aldana, Miguel Fuentes-Cabrera, Martín Zumaya

**Affiliations:** 1Instituto de Ciencias Físicas, Universidad Nacional Autónoma de México, Avenida Universidad s/n, Colonia Chamilpa, Cuernavaca 62210, Morelos, Mexico; martinzh@icf.unam.mx; 2Centro de Ciencias de la Complejidad, Universidad Nacional Autónoma de México, Coyoacán 04510, Mexico City, Mexico; 3Oak Ridge National Laboratory, Center for Nanophase Materials Sciences, Oak Ridge, TN 37831, USA; fuentescabma@ornl.gov; 4Oak Ridge National Laboratory, Computational Sciences and Engineering Division, Oak Ridge, TN 37831, USA

**Keywords:** self-assembly, self-propulsion, self-organization, polymerization

## Abstract

Self-assembly is a spontaneous process through which macroscopic structures are formed from basic microscopic constituents (e.g., molecules or colloids). By contrast, the formation of large biological molecules inside the cell (such as proteins or nucleic acids) is a process more akin to self-organization than to self-assembly, as it requires a constant supply of external energy. Recent studies have tried to merge self-assembly with self-organization by analyzing the assembly of self-propelled (or active) colloid-like particles whose motion is driven by a permanent source of energy. Here we present evidence that points to the fact that self-propulsion considerably enhances the assembly of polymers: self-propelled molecules are found to assemble faster into polymer-like structures than non self-propelled ones. The average polymer length increases towards a maximum as the self-propulsion force increases. Beyond this maximum, the average polymer length decreases due to the competition between bonding energy and disruptive forces that result from collisions. The assembly of active molecules might have promoted the formation of large pre-biotic polymers that could be the precursors of the informational polymers we observe nowadays.

## 1. Introduction

The formation of long biological polymers inside the cell (e.g., proteins and nucleic acids) is an out of equilibrium process that is carried out by a molecular machinery which, nowadays, is extremely complex. Such complexity is expected to have emerged gradually during the early stages of the origin of life [1,2]. However, even within a primitive scenario, it is still unclear what were the mechanisms that lead to the polymerization of large biological molecules from small monomers [3,4,5,6,7]. Recent work have shown that the assembly of large biological structures is considerably enhanced if it occurs far from equilibrium [8,9]. Two ways of introducing non-equilibrium conditions have been studied so far. The first one considers assembly in the presence of thermal gradients [8,9]. The second allows molecules to constantly move in a self-propelled manner due to the continuous supply of external energy [10,11]. The latter has been termed active-or dynamic self-assembly, and it is believed to have played an important role in the formation of large biological molecules from smaller constituents in pre-biotic Earth [12].

In this work we consider a model of active self-assembly in which self-propelled molecules, represented as a polygons, (see Figure 1), move in a thermal bath at temperature *T*. The polygonal shape is inspired by the shape of natural and unnatural DNA/RNA bases, which are know to form a variety of hydrogen bonded structures in equilibrium conditions. Further, when DNA/RNA bases and nucleosides are deposited on substrates, they are known to align along features of the substrates which, acting as sinks, attract molecules and promote the formation of chain-like structures [13,14,15,16,17]. We do not consider the effect of a substrate, but focus instead on studying how self-propelled molecules might assemble in non-equilibrium conditions. In this scenario, we assume that two molecules can either repel or bind, depending on the angle of the collision. The interaction between the molecules is defined through Lennard-Jones potentials, and the attractive and repulsive interactions are tuned by different cut-offs.

Numerical simulations clearly show that self-propulsion considerably speeds up the assembly of polymers, somewhat in agreement with previous related studies [10,11]. (Unlike [10,11], which considers only the formation of two dimensional structures, here we explicitly consider the formation of linear polymers.) Before proceeding to present the model used here and the results it produces, it is convenient to explain the differences between self-assembly and self-organization. We do this in the following section.

## 2. Self-Propulsion, Self-Assembly and Self-Organization

Self-assembly is a spontaneous process that takes place in equilibrium, where small dissociated blocks minimize the free energy of the system by forming macroscopic organized structures. This is the same process by which salt crystals, lipid membranes and other structures are formed [18,19,20,21]. Self-assembly is characterized by the fact that the dissociated blocks (molecules, colloids, etc.) move and bind together in equilibrium in a thermal bath at constant temperature *T*. They perform Brownian motion and are driven solely by the effect of temperature. The thermal motion leads to collisions, and the combination of attraction and repulsion (from collisions) results in the spontaneous formation of a variety of structures. An area of intense research in the last two decades has been the study of the self-assembly of *patchy colloids* [22,23,24]. These are molecules which have different attraction and repulsion zones (patches) around them and, depending on the geometry of the molecule and the distribution of its patches, these molecules can self-assemble in a variety of different structures, from linear chains to two-dimensional triangles to three-dimensional icosahedra [22]. In most of these studies it is assumed that the assembly of the patchy molecules occurs at (or near) thermodynamic equilibrium. However, *self-organization* is fundamentally different to *self-assembly* [8,25]. Self-organization requires a constant supply of external energy into the system for the constituent dissociated blocks to organize into ordered structures. Self-organized systems are, by definition, far from thermodynamic equilibrium and driven by external forces, not just by thermal motion.

Self-organization is prevalent at many scales in biology. Consider, for instance, the formation of macromolecules such as proteins, nucleic acids or ribosomes, inside the cell: all of them require the constant input of energy (via nutrients that are converted into ATP, GDP or enzymes). At a larger scale, self-organization is observed in the motion of flocks of birds, fish schools and bacterial colonies, to name a few [26,27]. At such scales, the constituent blocks (bird, fish, bacterium) are driven by the constant energy influx obtained from the intake of nutrients. It is apparent that the formation of structures in biology requires self-organization, not just self-assembly. It remains to be seen whether self-organization played a role in pre-biotic chemistry, e.g., in the formation of primordial informational oligomers. However, given the current prevalence of self-organization in biology, it seems reasonable to include non-equilibrium scenarios when investigating pre-biotic routes to macromolecular formation [5,28].

A first step to address this problem is to consider the assembly of self-propelled particles [11]. Systems of interacting self-propelled particles have been extensively studied [26,29], starting with the seminal paper by Vicsek et.al [29], where the so called Vicsek model of collective motion was introduced. In this model, self-propulsion consists in the motion of particles that maintain a constant speed, regardless of the interactions and forces acting on them. In the Vicsek model for a flock of birds, each bird moves with a constant speed, even when it interacts with other birds in the flock. These interactions only change the direction of motion of the bird, but not its speed. It is assumed that the energy necessary for each bird to sustain a constant speed is provided by its metabolism, i.e., the food each bird has to eat in order to be alive, move and fly. Ultimately, this food is an external source of energy that keeps the bird moving for long periods of time. The Vicsek model can be applied to any particle in general, not just birds, and each particle moves in the average direction of motion of the particles within a local neighborhood of radius $r_{0}$. To this average direction of motion, a white uncorrelated noise of amplitude η is added. The Vicsek model exhibits an order-disorder phase transition driven either by the noise intensity η for fixed particle density ρ, or by the density ρ for fixed noise intensity η. In the ordered phase, the system displays coherent motion, where all the particles move approximately in the same direction; in the disordered phase, particles move in random uncorrelated directions (like Brownian particles). One of the virtues of the Vicsek model is that simple local interaction rules can give rise to collective organized behavior in many-particle systems out of equilibrium. However, despite its apparent simplicity, the dynamics of the Vicsek model have not been thoroughly understood. In particular, whether the phase transition is first or second order, and the formation of traveling bands parallel to the boundaries, have been topics of intense research and discussion [30].

While self-propulsion makes sense for birds, fish and bacteria, its applicability to molecules is not completely justified. Nonetheless, there are experimental studies that have managed to include propulsion in molecular systems. One of the most significant ones is that of Bausch’s group [31,32]. This group has deposited micron size filaments on a substrate that was decorated with molecular motors. The motors threaded and subsequently pushed the filaments, providing a source of constant motion. The filaments were labeled with fluorophores, and their collective movement imaged. By changing the density, Bausch’s group revealed the formation of mesmerizing dynamic structures, e.g., vortices, moving clusters and moving bands, that have been considered and observed in previous studies.

Others have recognized that self-propulsion may have played an important role in the formation of pre-biotic molecules [12]. For instance, [11] showed that self-propelled colloids assemble faster and form larger structures than colloids without self-propulsion. Along the same lines, the formation of protein domains on the surface of the cell membrane was analyzed under two scenarios [8]. In the first scenario, the proteins interact, via electrostatic forces, with the phospholipids of the membrane. These forces drive the spontaneous assembly of the proteins through equilibrium processes, with the consequent formation of protein domains on the cell membrane. In the second scenario, the proteins, which are assumed to be close to the membrane, are out of equilibrium due to a phosphorylation-dephosphorylation cycle driven by ATP. The formation of protein domains on the cell membrane is also observed, however, these domains are considerably larger (about four hundred times fold) than those formed via equilibrium processes. These results show that the external input of energy, via self-propulsion or other sources (such as ATP), can considerably enhance the assembly of structures. To the best of our knowledge, formation of DNA-like polymers in non-equilibrium scenarios have not been considered yet. It’s very well known, however, that DNA-like bases deposited on substrates do form a variety of substrates, some of them linear, but all these studies have taken equilibrium conditions into account [13,14,15,16,17].

Here we take a first step towards investigating how self-propelled monomers might form polymeric-like structures in the absence of a substrate. For this we borrow ideas from recent studies where the assembly of self-propelled molecules or colloids was investigated [10,11,28,33,34]. In these studies, dissociated blocks were placed in a thermal bath at temperature *T*, and in addition to performing Brownian motion, they are thought to be able to transform energy from its surrounding into directed motion. The combination of repulsive and attractive interactions, as well as geometry, symmetry and direction of motion, lead to emergent collective behavior, such as collective motion, jamming, motility induced phase separation and the assembly of structures. An interesting example of this type of systems is the one presented in [33], where different organisms are modeled as self-propelled colloids composed of a collection of rigid spherical subunits. Similarly, others have investigated the non-equilibrium structures that emerge in a binary system of dipolar colloids that self-propelled in opposite directions, revealing the formation of chain and complex lane-like structures [35]. An excellent review of the type of structures formed by such colloids can be found in [36]. In the next section, we show how a model that borrows some of these ideas can be used to study polymer formation from a collection of self-propelled monomers.

## 3. Model Definition

Recent studies have shown that steric interactions and geometry are relevant for the emergence of different kinds of collective behavior in systems of self-propelled colloids. Rigid molecules should act in a similar manner, their steric interactions and geometry causing blockages that prevent motion. Therefore, for rigid molecules, the usual assumption of constant speed (as in the original Vicsek model) seems not longer adequate. Instead, it seems more adequate to assume a constant self-propulsion force acting on each rigid molecule. This requires the magnitude and direction of the velocity of each molecule to be modified according to the interactions it has with other molecules in the system. We will not attempt to provide a theoretical description of the active self-assembly process presented here. As mentioned in the previous section, even the simplest model of self-propelled particles, namely the Vicsek model, is already difficult enough not to be fully understood. Therefore, we will present only numerical results obtained through molecular dynamics simulations.

Our model consists of *N* active rigid molecules {1,2,…,N} characterized by the position of their center of mass $r_{i}$, velocity $v_{i}$ and orientation ${\hat{u}}_{i}=\left(\cos\theta_{i},\sin\theta_{i}\right)$. Following Wensink and Löwen [37], and Mallory [11], the sides of each molecule are discretized into *M* spherical subunits of radius $\sigma_{s}$ and positions $e_{i}^{\alpha }$, where the superscript refers to the α-th subunit, and the subscript refers to the *i*-th parent molecule (see Figure 1a). Subunits within the same molecule do not interact with each other, however, subunits of different molecules interact through Lennard-Jones (LJ) potentials of the form
(1)$$U\left(e_{ij}^{\alpha \beta }\right)=4\epsilon \left[{\left(\frac{\sigma_{s}}{e_{ij}^{\alpha \beta }}\right)}^{12}-{\left(\frac{\sigma_{s}}{e_{ij}^{\alpha \beta }}\right)}^{6}\right]$$
where $e_{ij}^{\alpha \beta }=e_{i}^{\alpha }-e_{j}^{\beta }$, is the relative distance between the α-th subunit of the *i*-th molecule and the β-th subunit of the *j*-th molecule. The total interaction potential $U_{ij}$ between the *i*-th and the *j*-th molecules is given by
(2)$$U_{ij}=\sum_{\alpha ,\beta }U\left(e_{ij}^{\alpha \beta }\right).$$

We consider a *dry* scenario, and therefore hydrodynamic interactions between molecules and the surrounding media are not taken into account. The dynamics of the system takes place on a 2D-plane and evolve according to the following pair of coupled over damped Langevin equations for the translational and rotational motion of each molecule:(3)$$\begin{array}{cc} \frac{\partial r_{i}}{\partial t} & =-\frac{1}{\gamma_{t}}\sum_{i\neq j}\nabla_{r_{i}}U_{ij}+F_{\mathrm{sp}}{\hat{u}}_{i}+\sqrt{2\gamma_{t}^{2}D}\xi_{i}^{t}\left(t\right),\\ \frac{\partial \theta_{i}}{\partial t} & =-\frac{1}{\gamma_{r}}\sum_{j\neq i}\nabla_{\theta_{i}}U_{ij}+\sqrt{2D_{r}}\xi_{i}^{r}\left(t\right), \end{array}$$
where $\gamma_{t}$ and $\gamma_{r}$ are the translational and rotational friction coefficients, $F_{\mathrm{sp}}$ is the magnitude of the self-propulsion force, and *D* y $D_{r}$ are the translational and rotational diffusion coefficients, respectively. $\xi^{t}$ and $\xi^{r}$ are Gaussian white noise terms for both the translational and rotational motion, and are given by $\langle \xi_{i}^{t}\left(t\right)\rangle =0$, $\langle \xi_{i}^{t}\left(t\right)\cdot \xi_{j}^{t}\left(t^{\prime }\right)\rangle =\delta_{ij}\delta \left(t-t^{\prime }\right)$, $\langle \xi_{i}^{r}\left(t\right)\rangle =0$ and $\langle \xi_{i}^{r}\left(t\right)\xi_{j}^{r}\left(t^{\prime }\right)\rangle =\delta_{ij}\delta \left(t-t^{\prime }\right)$.

Forces between molecules are obtained from the gradient of the interaction potential, $\nabla_{r_{i}}U_{ij}$, and the torques generated by these forces are obtained by differentiating the interaction potential with respect to their direction of motion $\theta_{i}$, i.e., $\nabla_{\theta }U_{ij}$. Equations (Equation 3) describe the motion of the center of mass (COM) and orientation of each molecule. All the spherical subunits within a molecule move and rotate rigidly with respect to the COM.

Interactions between the subunits in separate molecules can be either attractive or repulsive, depending on which side of the molecule they are located (see Figure 1b). Different cut-off distances, $r_{c}$, ensure that the interaction potential given by Equation (Equation 1) becomes zero when the two spherical subunits are separated by a distance larger than $r_{c}$. In the case of repulsive interactions, the cut-off distance is set to $r_{c}=2^{1/6}\sigma_{s}$; for the attractive case, $r_{c}=2.5\sigma_{s}$. In both cases, the potential is shifted after truncation so that $V(r_{c})=0$.

The geometry chosen for each molecule (Figure 1a) and the definition of the attractive and repulsive interactions, allow for one type of chain assembly only (see Figure 1c). This is of course a simplification, yet it is a first step towards understanding how self-propulsion affects polymerization. The parameter values used in the numerical simulations are given in Table A1.

## 4. Results

For systems composed of N=512 active molecules, numerical simulations were used to investigate two scenarios: (i) a system with periodic boundary conditions and (ii) a system with semi-periodic boundary conditions. In the first scenario, the molecules move inside a square box of sides *l* in which the four boundaries are periodic (Figure 2). In the second, the molecules move inside a rectangle of sides $l_{1}$ and $l_{2}$, with $l_{1}\leq l_{2}$. The two longest sides are rigid walls the molecules interact with through repulsive LJ potentials, whereas the two shortest sides are periodic, resembling a channel (Figure 3). We will refer to the first scenario as the bulk and to the second one as the channel. The motivation for studying the channel system is based on the hypothesis that the formation of organic macro-molecules relevant to the origin of life may have occurred in micro-channels formed either in meteorites [38,39,40,41], or in confined micro-spaces of marine vent chimneys in which molecules are driven by thermal gradients [42]. If the molecules are very small compared to the with of the channel, the bulk can be considered as the middle part of the channel in which the molecules do not interact with the confining walls. However, for high densities or for large molecules (compared to the width of the channel), the walls are expected to have a strong effect on the dynamics of the system. In what follows we present results from numerical simulations performed for both the bulk and the channel, and for different values of the thermal bath temperature *T* and the self-propulsion force $F_{\mathrm{sp}}$. In addition, for the channel, different values of the channel aspect ratio, defined as $R=l_{1}/l_{2}$, were considered.

Figure 2 shows different realizations for the bulk at constant temperature T=0.1 and different values of the self-propulsion force $F_{\mathrm{sp}}$. The case $F_{\mathrm{sp}}=0$ corresponds to spontaneous self-assembly, as it is driven only by thermal noise. It is apparent from Figure 2a that in this case, only very short polymers are formed while most of the molecules remain unbound. Contrary to this, for large values of the self-propulsion force (for instance $F_{\mathrm{sp}}=10$, Figure 2c), much longer polymers are created. Similar results are obtained for the channel, as Figure 3 shows. Interestingly, for the channel, most of the polymers tend to form and aggregate next to the repulsive walls, (see Figure 3). This is an unexpected result because there is no attractive force between the walls of the channel and the molecules. The walls were implemented to repel the molecules using the repulsive part of the LJ potential in the very same way as we implemented repulsion between the repulsive borders of the colloid molecules. Therefore, after a collision with the wall, the molecule can either bounce back into the bulk or slither along the wall. A similar aggregation next to the confining walls was also observed in two different systems of active colloidal rods [43] and spheres [44].

Figure 4 shows the average length of the polymer chain, 〈L(t)〉, as a function of simulation time *t* for different temperatures, *T*, self-propulsion forces, $F_{\mathrm{sp}}$, and aspect ratios, *R* (for the channel). The length *L* of a polymer in the system is defined as the number of monomers comprising it, and the average polymer length $\bar{L(t)}$
*in one realization* is computed over all the polymers at time *t*, i.e.,

Without losing sight of the limitation cited above, i.e., that our simulated systems might either be kinetically trapped or are still in a transient configuration moving towards the stationary state, we decided to investigate the dependence of the average chain length 〈L〉, and the chain length distribution P(L), on *T*, *R* and $F_{\mathrm{sp}}$ for the longest simulation times considered. All the results presented in what follows correspond to the maximum simulation time *t* = 10,000, which correspond to $10^{7}$ time steps of the numerical integration algorithm.

To quantify the effect that self-propulsion has on the polymerization process, we calculated, for different values of the self-propulsion force $F_{\mathrm{sp}}$ and temperature T=0.1, the probability P(L) that a polymer of length *L* is created. The results are shown in Figure 5a for the system with periodic boundary conditions. It is clear that by increasing the magnitude of $F_{\mathrm{sp}}$, longer polymers are created. However, this behavior is not monotonic, as seen in Figure 5b, which shows the average polymer length 〈L〉 as a function of $F_{\mathrm{sp}}$ for different temperatures. It is apparent from Figure 5b that at low temperatures (T≤0.5) there exists an optimal value, $F_{\mathrm{sp}}^{⋆}$, of the self-propulsion force for which 〈L〉 reaches a maximum. When $F_{\mathrm{sp}}$ increases beyond this optimal value, the average polymer length 〈L〉 starts to decrease. This is due to the molecule-polymer and polymer-polymer collisions. When $F_{\mathrm{sp}}$ is too large, these collisions are very energetic, disassembling the already formed polymers. Figure 5c shows that the optimal value $F_{\mathrm{sp}}^{*}$ decreases with temperature in the low-temperature regime. However, at high temperatures (T=0.75) there is no optimal value of $F_{\mathrm{sp}}$ at which 〈L〉 maximizes, as Figure 5d shows. Actually, from this figure it is apparent that, in the high-temperature regime, the system reaches a stationary state within the simulation time. Note also that the stationary value of 〈L〉 monotonically increases with $F_{\mathrm{sp}}$. Finally, the inset in Figure 5d clearly shows that at high temperatures mostly monomers and dimers coexist, whereas longer polymers appear in negligible amounts.
(4)$$\bar{L(t)}=\frac{1}{N_{p}\left(t\right)}\sum_{i=1}^{N_{p}\left(t\right)}L_{i},$$
where $N_{p}\left(t\right)$ is the number of polymers in the system and $L_{i}$ is the length of the *i*-th polymer. Then, we compute the ensemble average 〈L(t)〉 by performing *M* realizations of the system with the same parameters and averaging $\bar{L(t)}$ over the *M* realizations. In what follows, we present results for M=20 realizations.

In every case reported in Figure 4, it can be seen that increasing $F_{\mathrm{sp}}$ increases the growth rate of the polymer chains (the slope of the curve increases with $F_{\mathrm{sp}}$). It should be noted that in none of these cases, the system has been simulated long enough to reach a steady-state. Nonetheless, it seems clear that even during the transient time, adding a self-propulsion force speeds up the rate at which the chains grow.

Similar results are presented in Figure 6 for the channel system. The tail of the distribution P(L) (inset in Figure 6a) reveals that much longer polymers are created in the channel than in the bulk (compare with the inset in Figure 5a). This is due to the aggregation of molecules and polymers next to the repulsive channel walls, which decreases the velocity of the molecules and polymers preventing the occurrence of very energetic collisions which may disassemble the already formed polymers. In fact, from Figure 6b it can be observed that at low temperatures (T=0.1) the average polymer length 〈L〉 for the channel system does not reach a maximum at any value of the self-propulsion force $F_{\mathrm{sp}}$. Instead, 〈L〉 monotonously increases with $F_{\mathrm{sp}}$ and then saturates. However, at higher temperatures (T=0.25) 〈L〉 behaves again non-monotonously with respect to $F_{\mathrm{sp}}$, reaching a maximum at an optimal value $F_{\mathrm{sp}}^{⋆}$ of the self-propulsion force, as Figure 6c shows. This optimal value seems to increase with the channel aspect ratio *R* (see Figure 6d), which indicates that the narrower the channel, the less self-propulsion force is needed to create long polymers. This again reflects the effect of the repulsive walls in the formation of polymers. This effect clearly becomes stronger for narrow channels.

In order to compare the dynamics of the systems with and without confining walls, in Figure 7 we plot, for each value of $F_{\mathrm{sp}}$, the average maximum polymer length $\langle L_{max}^{C}\rangle$ observed in the channel against the average maximum polymer length $\langle L_{max}^{B}\rangle$ observed in the system with periodic boundary conditions (the bulk), for two different temperatures and different values of the channel aspect ratio *R*. To generate the data reported Figure 7 we focused on the longest polymers in each realization instead of just computing the average polymer length over all the polymers. From this figure it is clear that almost all the points fall quite above the identity line (except for $F_{\mathrm{sp}}=0$). This means that the confining walls of the channel do enhance the formation of longer polymers with respect to the bulk. The aggregation and jammingof active particles next to the confining walls has been observed in both numerical simulations and experiments and it is still an open problem to determine why this phenomenon happens [43,44,45,46,47].

## 5. Conclusions

We have presented a polymerization model of active self-propelled molecules. Our results show that self-propulsion considerably speeds up the polymerization of these molecules, as it increases the rate at which a chain grows. As the magnitude of the self-propulsion force increases, longer polymers appear in comparison with the equilibrium case in which the spontaneous self-assembly is driven only by the Brownian motion of molecules as a result of the temperature of the thermal bath. When the molecules move in “free space” (fully periodic boundary conditions), there is an optimum value of the magnitude of the self-propulsion force for which the mean polymer length is maximum, and above which it decreases. This is due to the competition between the bonding energy and the force exerted between molecules in collisions, where the repelling subunits interact. This competition takes place less often when the molecules move inside a channel with repulsive walls. In this case, the molecules and polymers tend to aggregate next to the walls, which decreases the velocity of both, molecules and polymers, preventing the occurrence, to some extent, of highly energetic collisions. Therefore, confinement inside repulsive boundaries also enhances the polymerization process. This is apparent from Figure 7, which shows that in the channel the polymers are considerably longer than in the bulk. A question, however, remains open: how much are the results observed affected by the simulation time? From Figure 4 it is clear that the simulation time considered ($10^{7}$ time steps) does not seem to be enough for the system to reach a stationary state. Therefore, the results presented here could be a consequence of kinetic effects and longer simulation times could produce chain length distributions P(L) that are independent of the self-propulsion force. However, the results presented in Figure 5d suggest otherwise. For the case reported in this figure, which correspond to the bulk in the high temperature regime, the system seemed to have reached the stationary state, and the stationary value of 〈L〉 does increase with the self-propulsion force $F_{\mathrm{sp}}$. While this issue will be investigated in future studies, for now it seems safe to conclude that self-propulsion does speeds up the rate of polymerization.

The model we have introduced takes into account the full geometry of the molecules by discretizing them into subunits. Due to the definition of the interactions between molecules and the assembly direction induced by their geometry, one could think of a similar model based on patchy colloids [24,48,49,50] or dipolar particles [35,36], in which the attractive and repulsive regions could be modeled as patches decorating the surface of rigid disks. Nonetheless this approach would not take into account the geometry of the molecule but a simplified version of it, and even if patchy colloids are able to self-assemble into chains [50,51] or more complex structures [52], it is not clear the equivalence of both models.

Our results, and the ones obtained by other groups, show that self-propulsion acts as a catalyst in the assembly of ordered macroscopic structures from small molecules, let them be one-dimensional polymers, as in the model presented in this work, or capsides [11] or protein domains in a cell membrane [8]. Altogether, these results suggest that self-propulsion could have been an important mechanism for the formation of pre-biotic structures or the synthesis of nanometric structures. More theoretical and experimental work, however, is needed to shed light on this claim.

## Figures and Tables

**Figure 1 entropy-22-00251-f001:**
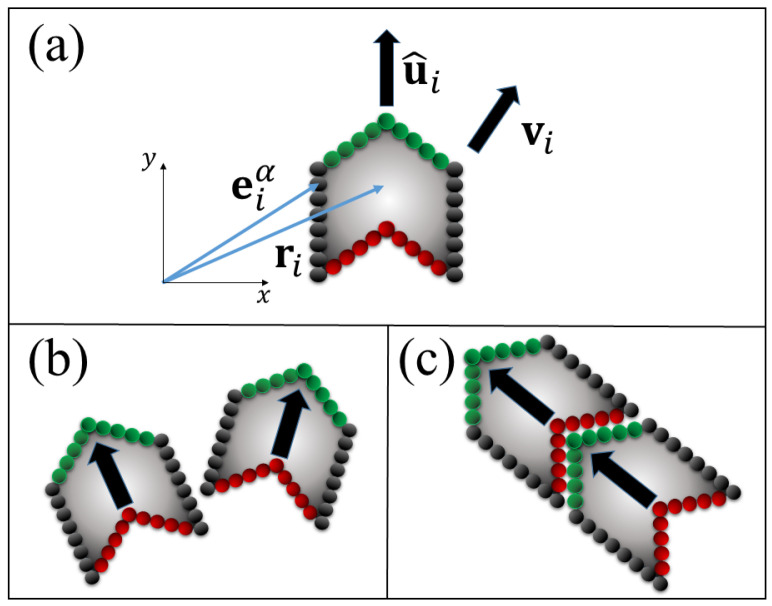
(**a**) Schematic representation of the basic rigid molecule. The sides of the molecule are discretized into spherical subunits with positions $e_{i}^{\alpha }$ which exert attractive or repulsive forces on the spherical subunits of other molecules. The entire molecule is characterized by the position $r_{i}$ of the center of mass, its velocity $v_{i}$ and orientation ${\hat{u}}_{i}$. Note that the velocity and orientation are not necessarily parallel. The self-propulsion force always acts in the direction of the orientation vector ${\hat{u}}_{i}$. The edges of the molecule are colored according to the repulsive or attractive forces they exert on other molecules: if two molecules collide through edges of the same color, the force is repulsive as in (**b**), whereas the force is attractive only between the green and red edges, as in (**c**). The dynamics take place on a 2D-space.

**Figure 2 entropy-22-00251-f002:**
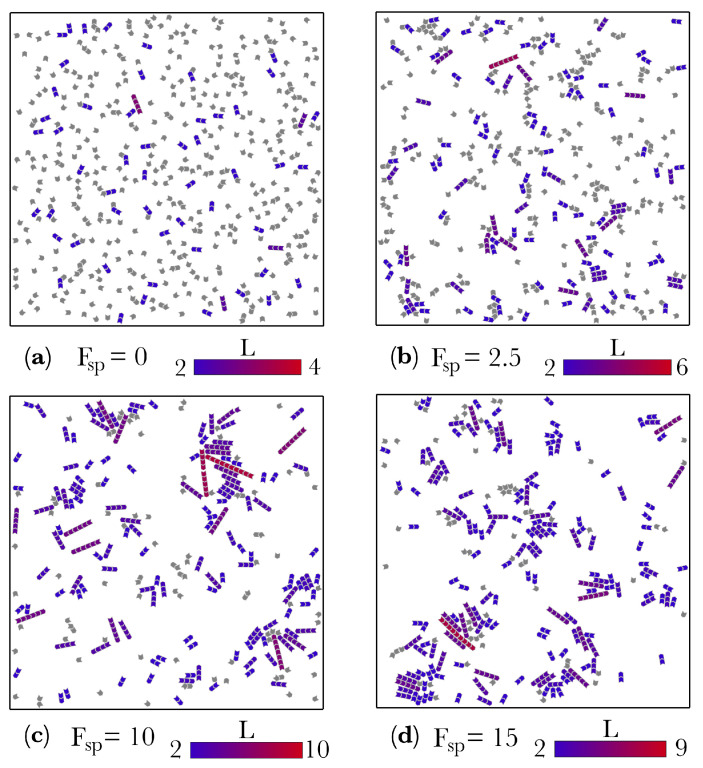
Snapshots of the system with fully periodic boundary conditions (the *bulk*) showing the effect of self-propulsion on the assembly of molecules. Different snapshots correspond to different values of the self-propulsion force: (**a**) $F_{\mathrm{sp}=0}$, (**b**) $F_{\mathrm{sp}=2.5}$, (**c**) $F_{\mathrm{sp}=10}$, and (**d**) $F_{\mathrm{sp}=15}$. In all cases the temperature is T=0.1 and the number of particles is N=512. Note that with no self-propulsion (panel (**a**), $F_{\mathrm{sp}}=0$), only very small chains are formed. This case would correspond to spontaneous self-assembly. As the self-propulsion force $F_{\mathrm{sp}}$ increases, (panels (**b**–**d**)), longer chains appear. It is apparent that self-propulsion considerably improves the assembly of molecules as compared to spontaneous self-assembly. The color gradient represents the length of the chain. All the monomers (L=1) are colored in gray.

**Figure 3 entropy-22-00251-f003:**
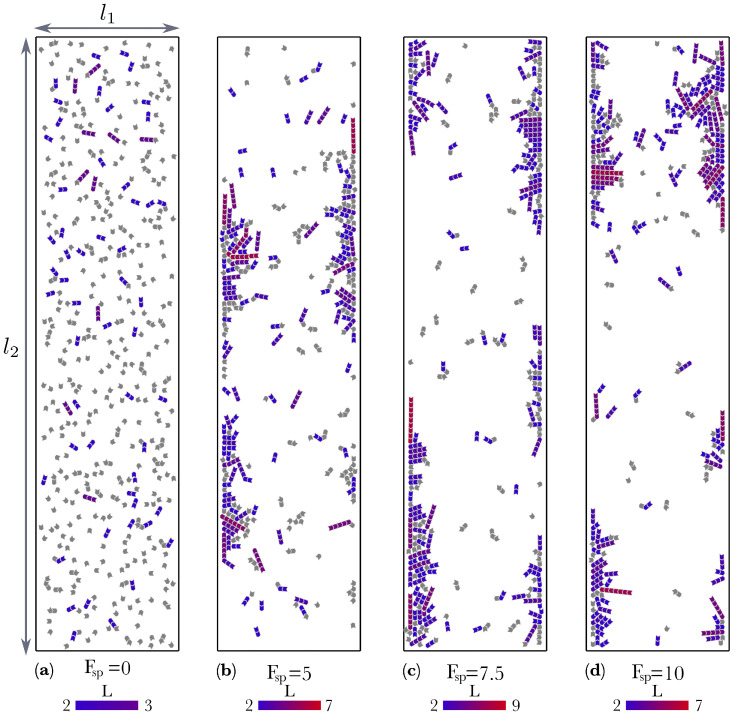
Effect of self-propulsion on the assembly of molecules within semi-periodic boundary conditions (the *channel*), where the longest sides are rigid boundaries and the shorter ones are periodic. All the snapshots correspond to temperature T=0.1, channel aspect ratio $R=l_{1}/l_{2}=0.25$ and N=512 particles. (**a**) For $F_{\mathrm{sp}}=0$, i.e., when the motion of the molecules is driven just by thermal noise, only small polymers are formed. The next three panels correspond to non-zero values of the self-propulsion force: (**b**) $F_{\mathrm{sp}}=5$, (**c**) $F_{\mathrm{sp}}=7.5$, and (**d**) $F_{\mathrm{sp}}=10$. Note that as the value of $F_{\mathrm{sp}}$ increases the length of the assembled polymers also increases. Note also that in this case the polymers aggregate in the repulsive boundaries, allowing them to form even longer chains. The color gradient represents the length of the chain. All the monomers (L=1) are colored in gray.

**Figure 4 entropy-22-00251-f004:**
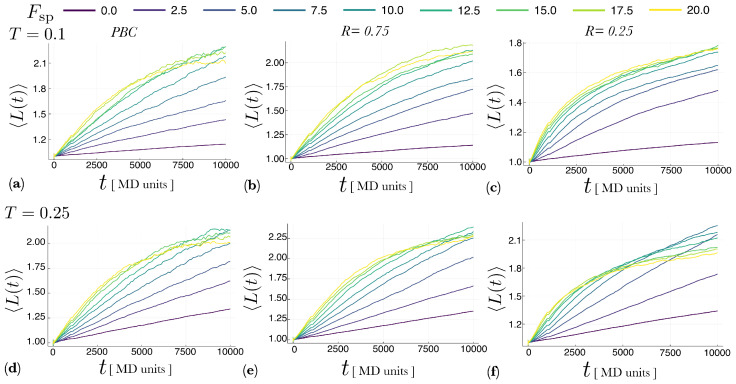
Average length 〈L〉 of polymer chains as a function of time *t* for different values of the aspect ratio *R*, temperature *T* and self-propulsion force $F_{\mathrm{sp}}$. Panels (**a**–**c**) correspond to temperature T=0.1 and panels (**d**–**f**) to T=0.25. Panels (a,d) show the evolution of 〈L(t)〉 in the case of periodic boundary conditions, whereas the remaining panels correspond to the channel geometry for different aspect ratios: in (**b**,**e**) R=0.75 and in (**c**,**f**) R=0.25. It can be observed that during the computing time 〈L〉 does not reach a stationary value. However, it is evident in all cases that increasing the self-propulsion force $F_{\mathrm{sp}}$ speeds up the formation of polymers. The maximum computing time t=10,000 corresponds to $10^{7}$ time steps in the molecular dynamics simulation. For each curve, the ensemble average 〈L〉 was computed over 20 different realizations.

**Figure 5 entropy-22-00251-f005:**
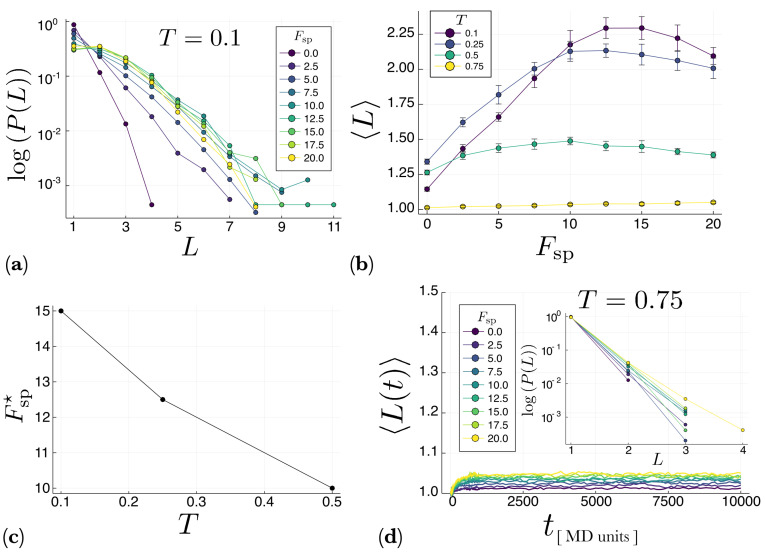
Results for the system with periodic boundary conditions. (**a**) Probability P(L) that a polymer of length *L* is created after $10^{7}$ time steps. Different colors correspond to different values of the self-propulsion force $F_{\mathrm{sp}}$. All the simulations in this panel were computed at constant temperature T=0.1. The inset shows a magnification of the tail of the distribution P(L) in order to better appreciate the existence of long polymers. (**b**) Average chain length 〈L〉 as a function of the self-propelled force $F_{\mathrm{sp}}$ for different values of the temperature *T*. Note that for lower temperatures, there is an optimal value of $F_{\mathrm{sp}}^{⋆}$ of the self-propulsion force at which 〈L〉 exhibits a maximum. This value is plotted in panel (**c**) as a function of the temperature *T*. (**d**) Temporal evolution of the average polymer length, 〈L(t)〉, in the high-temperature regime for different values of $F_{\mathrm{sp}}$. Note that in this case 〈L(t)〉 does reach a stationary value which increases with $F_{\mathrm{sp}}$. The inset shows the fraction of monomers (L=1) and dimers (L=2) in the system in this regime. (Polymers with L>2 appear in such low quantities that cannot be appreciated on the histogram).

**Figure 6 entropy-22-00251-f006:**
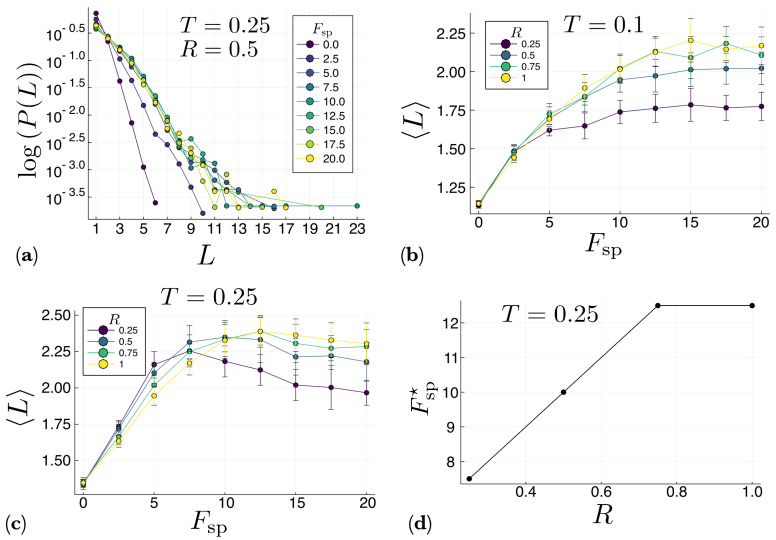
Results for the channel with semi-periodic boundary conditions. All the results presented in this figure were computed at constant temperature T=0.1. (**a**) Probability P(L) that a chain of length *L* is formed after $10^{7}$ time steps, for different values of the self-propulsion force $F_{\mathrm{sp}}$ in a channel with an aspect-ratio R=0.75. The inset shows a magnification of the tail of the distribution P(L). Note that in this case, even longer polymers are formed with respect to the case with fully periodic boundary conditions depicted in Figure 5. Panels (**b**,**c**) show the average chain length 〈L〉 as a function of the self-propulsion force $F_{\mathrm{sp}}$ for different values of the channel aspect-ratio *R*, and temperatures T=0.1 and T=0.25, respectively. For low temperature (T=0.1), the average chain length 〈L〉 first increases and then saturates as a function of $F_{\mathrm{sp}}$, whereas at a higher temperature (T=0.25) it reaches a maximum at an optimal value $F_{\mathrm{sp}}^{⋆}$. (**d**) Optimal value $F_{\mathrm{sp}}^{⋆}$ as a function of the channel aspect ratio *R* for T=0.25.

**Figure 7 entropy-22-00251-f007:**
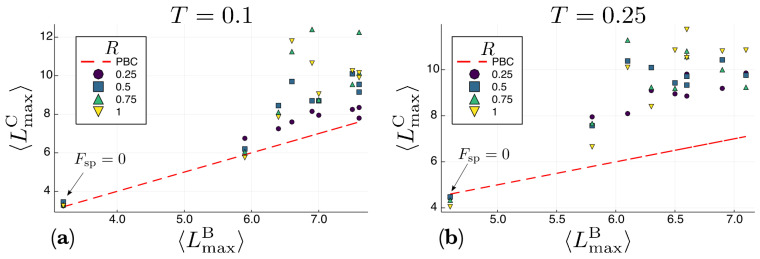
Average maximum polymer length $\langle L_{\max}^{C}\rangle$ for the channel against the average maximum polymer length $\langle L_{\max}^{B}\rangle$ for the bulk. Each point corresponds to a specific value of $F_{\mathrm{sp}}$, whereas the different symbols correspond to different values of the channel aspect ratio *R*. Panel (**a**) shows data for T=0.1 whereas panel (**b**) shows similar data at a higher temperature T=0.25. Note that almost all the points (except for $F_{\mathrm{sp}}=0$) fall above the identity line, which means that the polymers formed in the channel are considerably longer than the ones observed in the bulk. The ensemble averages 〈·〉 were computed over 20 different realizations for each condition.

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
