# Peer review of "Self-Propulsion Enhances Polymerization"

_entropy, 2020, doi:10.3390/e22020251_

Round 1
Reviewer 1 Report
The authors study the self-assembly (or self-organizing) properties of arrowheads-shaped colloids, finding that activity promotes the formation of linear chains, that are longer that the ones found in equilibrium.
I find the topic potentially very interesting and very nicely introduced (with a minor note); the methods used are also well described (with minor notes); the paper is also well written. Unfortunately I have two major issues with this paper:
1. The paper completely neglects the results on the self-assembly of patchy colloids. Indeed the model proposed is patchy-like: the particles are attracted to each other and form bonds only if they face in a certain direction. To me it could be mapped (without activity) either to a 2-patch colloid with patches of different kind or even to a dipolar particle (see the works of L. Rovigatti, and F.Sciortino on the topic). Clearly such models (depending on the temperature and concentration) have, in the bulk, very specific ground states, which feature relatively long self-assembled chains (certainly more than 1 particle long). Further, in view of the "dipolar particles" analogy, two relevant papers that are not cited are
F. Kogler and S. H. L. Klapp 2015 EPL110 10004
S.H.L. Klapp Current Opinion in Colloid & Interface Science 21 (2016) 76–85
2. The other main concern is about the "steady state". The point is that the system chosen is tricky: the colloid is very damped (I guess for technical reasons) and has an overall radius of ~2-3 units. Thus, in the passive case and at the temperature chosen, the time the colloid needs to diffuse around an area that is of the order of its size^2 is given by
tau = R^2/4D = 4/4*(0.1/250) = 2500 (5600 for R=3),
which means (with the time step chosen) order 10^6 steps, which is very close to the 10^7 steps reported in table A1. This would mean that particles have diffused very little and indeed very few encounters have taken place. Even in the case that the number reported in Table A1 would be in MD units (thus 10^10 steps) I argue it would be enough to really reach the equilibrium self-assembly scenario. Do not forget that as particle assemble, the self-assembly kinetics slows down because of the size of the cluster as clusters are slow and difficult to merge. Additionally, for linear chains, the assembly is slowed by the fact that the only way to access the self-assembled cluster is through its ends.
The point I want to make is the following: the message of the paper is that the formation of long chain is enhanced by activity but the data only show, to me, that the enhancement is a kinetic effect due to sampling the system on a short time window. Indeed, in practice, it is impossible to properly reach the final self-assembled scenario at such temperatures in a Molecular Dynamics simulation. This latter fact is well known to people that perform simulations of these systems, and indeed they employ Monte Carlo techniques and often use ad-hoc Monte Carlo moves and enhanced sampling techniques (notice that, on the contrary, Nature had plenty of times back then, so she was not concerned with kinetics!).
Thus my real question would be: how did the author prove the "steady state" condition - or equilibration in the case of the passive system? Given the (almost) impossibility of carrying the simulation until the end, I would suggest the authors to emphasize the fact that the enhancement is a kinetic effect as much as possible.
Other notes:
1. I like the introduction very much, but the reference to the Vicsek model does not fit very well, as it is not used in the simulation. I would rather remove it or give it less emphasis (eg. removing fig. 2).
2. Further, when summarizing the Vicsek model, you should not use the same name (rho) for the density of obstacles and the density of the particles.
3. In figure 3, the two red lines spanning the panels are confusing, I would limit the left one to the first and second panel and the right one to the last panel. In truth, I would actually remove the figure altogether, it's not needed.
4. If Fig. 6b and 7b, the title of the legend should be "T" as in temperature, or? and what is "R"?
5. I would revisit fig. 6a and 7a, they are not very readable. Why not using a normal plot with symbols and lines (maybe on a semi-log scale)?
6. I would also advise to characterize the system a bit more, perhaps looking at the clusters formed even if they are just a bunch of particles collided (and maybe compare the clustering properties with the ones of an Active Brownian Particles system).
For the reasons above, I can't recommend the publication of this paper in the present state.
Author Response
Please, see the attached file.

Reviewer 2 Report
The manuscript itself is well-written, scientifically sound, and addresses an interesting and important area within non-equilbrium self-assembly and self-organization. This work provides a meaningful extension to understanding the self-assembly behavior of colloidal particles that undergo directed motion or self-propulsion. This is the first manuscript that I am aware of where the polymerization of self-propelled colloidial monomers are studied using a minimal model. The results will have immediate applications in optimizing material design and self-assembly, but also help to forge an understanding of the self-organization of micro-components in complex biological environments. The authors' work using this simple model system is a useful step in giving orientation into this very complex field. It will be interesting to see how the behavior of these "polymeric" systems is altered when additional physics are incorporated into the model and whether this behavior will manifest itself in three dimensional system. The results and phenomenology of this study are quite interesting and consistent with other similar studies for compact structures. It would be useful to the community if an analytical theory could be devised to quantitatively describe the observed behavior. In closing, the manuscript is timely and should be accessible and interesting to a broad group of readers.
The following papers may be of interest to the authors, but in no way need to be cited, as they also address the topic of active self-assembly and they may find some of the models implemented useful in future studies:
Mallory, Stewart A., and Angelo Cacciuto. "Activity-enhanced self-assembly of a colloidal kagome lattice." Journal of the American Chemical Society 141.6 (2019): 2500-2507.
Mallory, Stewart A., Chantal Valeriani, and Angelo Cacciuto. "An active approach to colloidal self-assembly." Annual review of physical chemistry 69 (2018): 59-79.
Omar, Ahmad K., et al. "Swimming to Stability: Structural and Dynamical Control via Active Doping." ACS nano 13.1 (2018): 560-572.
Mallory, S. A., et al. "Self-assembly of active amphiphilic Janus particles." New Journal of Physics 19.12 (2017): 125014.
Author Response
We sincerely thank the reviewer for her/his useful comments. We consider the references suggested by the reviewer relevant and so have been included in the text. Additionally, we have changed the manuscript to implement the suggestions of the other two reviewers. The changes in the revised version of the manuscript are colored in blue so that they are easy to identify within the text.
Reviewer 3 Report
The authors present a simple model to characterize the polymerization of active particles in two dimensions (considering the bulk case and the system confined between walls).
The model is based on rigid particles with a geometry that allows them to pile up into linear chains, and pushed by an active force acting on the center of mass.
By means of molecular dynamic simulations the author characterize the probability to observe an assembled polymer of given length, depending on the strength of the active force and on the temperature.
They found that, for the bulk system, there is an optimal strength for any explored temperature, leading to the average maximum length of the assembled polymers. For stronger active forces the energy of the collision disrupts the assembled structures. On the contrary, these phenomenon is not observed when the system is confined, as the collision with the wall largely reduce the velocity of the particles.
Overall the paper is fluent and clear, and I think it could be interesting in the community of soft matter and material science. For this reason, I'm inclined to support the publication. Nevertheless, a series of question should be addressed.
1) First of all, I think the analysis of the results could be improved, adding more data and comparison. For example,
1.1 ) with respect to Fig. 6, an other panel where the locus of maximum polymerization in the plane F_sp - T could help the visualization of the data and the discussion of the results
1.2 ) a comparison between the outcomes of the bulk simulations and confined simulations also could help. In particular, comparing <L> vs F_sp in confinement for different ratios between the sizes, with respect to what observed in the bulk. I would expect that when the ratio is 1, and when the finite size effect are negligible, the values of <L> should be comparable. It not this could be commented
1.3 ) It would be interesting to compute the average time an assembled particle needs to escape from the chain, for both systems and compare them. It probably increases with F_sp and decrease with T.
1.4) With respect to the previous point, also computing the mean square displacement MSD of the assembled polymers, and try to extract a diffusion coefficient (assumed that there is a regime where MSD~t) would enrich the discussion.
2) The author observe the formation of linear active chains, and with this respect, I think that the manuscript would benefit by citing the following references:
a) How does a flexible chain of active particles swell? Andreas Kaiser, Sonja Babel, Borge ten Hagen, Christian von Ferber, and Hartmut Löwen, J Chem Phys 142, 124905 (2015)
b) Self-propelled worm-like filaments: spontaneous spiral formation, structure, and dynamics, Rolf E. Isele-Holder, Jens Elgeti and Gerhard Gompper, Soft Matter, 11, 7181, (2015)
c) Globulelike Conformation and Enhanced Diffusion of Active Polymers, V. Bianco, E. Locatelli, and P. Malgaretti, Phys. Rev. Lett. 121, 217802 (2018)
d) Deviations from Blob Scaling Theory for Active Brownian Filaments Confined Within Cavities, S. Das and A. Cacciuto, Phys. Rev. Lett. 123, 087802 (2019)
In particular, the paper a) describe the structural and dynamical properties of polymers composed by active particles, where the vector of the active force freely diffuses (i.e. is not coupled with the polymer backbone); the paper b) focuses on the structural properties of active filaments, where the activity is tangent to the backbone, observing the formation of spirals; the paper c) creates a connection between the dynamical and structural properties of active polymers according to the coupling of the active force: whenever the activity is correlated to the backbone than the polymer undergoes to a globule transition and the diffusion coefficient turn to be independent on the polymer size; the paper d) focuses on the effect due to spherical confinement.
The work of the author would lead to polymer with an activity tangent to the polymer backbone, as described in papers b), c) and d). It is true than, within the author observe short polymer chains (essentially rigid by looking and the snapshot reported in the manuscript), but in principle longer chains could emerge by considering larger particle concentrations. These chains would have a persistence length and, as long as the chain does not disassemble, what observed in the manuscripts b) c) and d) could be observed.
3) I assume that the assembled polymers are rigid. Do the author observe any degree of curvature ? With this respect, it is not clear to me how the author compute <L>. Is it the end-to-end distance averaged over all the polymers and single particles, or the average number of assembled monomers ? This should be stated in the manuscript.
4) It is not clear to me how the author implement the bounce back condition of the active particles in presence of the confining wall. It has been observed that, according to this interaction, the behavior of active particles can be very different: they can bounce to the bulk or slither along the wall.
See, for example
a) Wall accumulation of bacteria with different motility patterns Paolo Sartori, Enrico Chiarello, Gaurav Jayaswal, Matteo Pierno, Giampaolo Mistura, Paola Brun, Adriano Tiribocchi, and Enzo Orlandini
Phys. Rev. E 97, 022610 (2018)
b) K. Drescher, J. Dunkel, L. H. Cisneros, S. Ganguly, and R. E.Goldstein, Fluid dynamics and noise in bacterial cell–cell andcell–surface scattering,Proc. Natl. Acad. Sci. USA108,10940 (2011)
c) M. Contino, E. Lushi, I. Tuval, V. Kantsler, and M. Polin, Microalgae Scatter off Solid Surfaces by Hydrodynamic and Contact Forces, Phys. Rev. Lett.115, 258102 (2015)
d) G. Li and J. X. Tang, Accumulation of Microswimmers Near a Surface Mediated by Collision and Rotational Brownian Motion, Phys. Rev. Lett.103,078101 (2009).
5) In Fig. 2 it is not clear to me if panels (b) and (c) refer to some simulation, if they are adapted from some reference or if are just qualitative to show the behavior of the system. This should be clarified
6) at line 41, please correct "we we" with "we"
7) at line 129, please correct "it is seems" with "it seems"
8) at line 142, please correct "take place" with "takes place"
9) Fig. 3 is not relevant, since it is just the plot of the Standard Lennard Jones potential (truncated at various distances) and could be removed.
10) To implement the move of the rigid molecule, do the author adopt a particular algorithm, or they just rigidly shift all the monomer particles according to the motion of the center of mass? Do the author check that the shape of the monomer is preserved, as usually rigid move introduce numerical errors ( due to unavoidable approximations done by the CPU) that accumulate over repeated time steps.
11) In Fig. 4 and Fig. 5 I guess that different colors represent chains of different length. This should be stated in the caption, and possibly a color code bar on the side of the picture could help.
Author Response
Please, see the attached file.

Round 2
Reviewer 1 Report
I thank the authors for the changes they have made to the manuscript, I find it now much improved and I suggest its publication.
A few minor comments:
1. I feel I have been, in part, misunderstood: I did not mean, with my comments, to say anything quantitative about the polymerization length at the steady state in presence of activity. My comment was mainly about the fact that also the passive system polymerizes, given time. That said, I like very much all the changes made, except the sentence at line 222 "It could very
well be that...": I find the sentence a bit out of the blue and potentially confusing; indeed it might not be the case. I would suggest to remove it and pick up from "it seems clear that... "
2. I would suggest, for the sake of readability, to make the axis labels bigger, where spacing allows. Further, I would say, in the axis titles where "t" appears, that "t" is in MD units. ( I mean "t [MD units]" or "t/tau_MD")
2. I still think that the histogram representation in 5a and 6a is still not ideal and I would definitely advise to present the data in a different way (I find the graph in the response letter fine), but I will not force you to do it.
Author Response
We thank you again for carefully reading the revised version of our manuscript. We have implemented the three changes you suggested.
Here we provide a point-by-point response to your comments:
1.- Your remark about the fact that polymerization in the passive system also might occur if we were to let the system evolve long enough, made us investigate whether or not the system had reached the stationary state not only in the passive case but also in the active cases. We really thank you for asking this question which opens up the possibility for a deeper investigation.
We removed the sentence starting with "It could very well be that..." as you suggested. This was the only change in the main text.
2.- We have increased the size of the axis labels and numbers in all the figures. We also replaced the label "t" by the label "t_[MD units]" in all the graphs that show the temporal evolution of the system.
3.- We changed the histograms by lines. We trust your judgment.
We thank you again for your valuable comments and suggestions. We sincerely believe that the presentation of the data was considerably improved thanks to your comments.